# DeepThal: A Deep Learning-Based Framework for the Large-Scale Prediction of the α^+^-Thalassemia Trait Using Red Blood Cell Parameters

**DOI:** 10.3390/jcm11216305

**Published:** 2022-10-26

**Authors:** Krittaya Phirom, Phasit Charoenkwan, Watshara Shoombuatong, Pimlak Charoenkwan, Supatra Sirichotiyakul, Theera Tongsong

**Affiliations:** 1Department of Obstetrics and Gynecology, Faculty of Medicine, Chiang Mai University, Chiang Mai 50200, Thailand; 2Modern Management and Information Technology, College of Arts, Media and Technology, Chiang Mai University, Chiang Mai 50200, Thailand; 3Center of Data Mining and Biomedical Informatics, Faculty of Medical Technology, Mahidol University, Bangkok 10700, Thailand; 4Department of Pediatrics, Faculty of Medicine, Chiang Mai University, Chiang Mai 50200, Thailand; 5Thalassemia and Hematology Center, Faculty of Medicine, Chiang Mai University, Chiang Mai 50200, Thailand

**Keywords:** alpha plus-thalassemia, hemoglobin H disease, machine learning, red blood cell indices, screening

## Abstract

Objectives: To develop a machine learning (ML)-based framework using red blood cell (RBC) parameters for the prediction of the α^+^-thalassemia trait (α^+^-thal trait) and to compare the diagnostic performance with a conventional method using a single RBC parameter or a combination of RBC parameters. Methods: A retrospective study was conducted on possible couples at risk for fetus with hemoglobin H (Hb H disease). Subjects with molecularly confirmed normal status (not thalassemia), α^+^-thal trait, and two-allele α-thalassemia mutation were included. Clinical parameters (age and gender) and RBC parameters (Hb, Hct, MCV, MCH, MCHC, RDW, and RBC count) obtained from their antenatal thalassemia screen were retrieved and analyzed using a machine learning (ML)-based framework and a conventional method. The performance of α^+^-thal trait prediction was evaluated. Results: In total, 594 cases (female/male: 330/264, mean age: 29.7 ± 6.6 years) were included in the analysis. There were 229 normal controls, 160 cases with the α^+^-thalassemia trait, and 205 cases in the two-allele α-thalassemia mutation category, respectively. The ML-derived model improved the diagnostic performance, giving a sensitivity of 80% and specificity of 81%. The experimental results indicated that DeepThal achieved a better performance compared with other ML-based methods in terms of the independent test dataset, with an accuracy of 80.77%, sensitivity of 70.59%, and the Matthews correlation coefficient (MCC) of 0.608. Of all the red blood cell parameters, MCH < 28.95 pg as a single parameter had the highest performance in predicting the α^+^-thal trait with the AUC of 0.857 and 95% CI of 0.816–0.899. The combination model derived from the binary logistic regression analysis exhibited improved performance with the AUC of 0.868 and 95% CI of 0.830–0.906, giving a sensitivity of 80.1% and specificity of 75.1%. Conclusions: The performance of DeepThal in terms of the independent test dataset is sufficient to demonstrate that DeepThal is capable of accurately predicting the α^+^-thal trait. It is anticipated that DeepThal will be a useful tool for the scientific community in the large-scale prediction of the α^+^-thal trait.

## 1. Introduction

Thalassemia is one of the most common inherited autosomal recessive diseases, especially in Southeast Asia. In Thailand, the prevalence of thalassemia traits is as high as 30%, and the prevalence of thalassemia disease is approximately 1% [1]. Thus, we have about 12,000 new cases per year, or about 48,000 couples at risk per year [2]. Homozygous α^0^-thalassemia (Hb Barts) involving deletion of all the four loci of the alpha-globin genes (*HBA2* and *HBA1*, genotype --/--) on chromosome 16 is usually lethal, ending up with hydrops fetalis and stillbirth due to the absence of the alpha-globin chain production. Hemoglobin H (Hb H) disease is the most common form of thalassemia intermedia with variable severity. In the majority of cases, Hb H disease results from deletions of three out of the four functioning HBA alleles, compound heterozygosity of α^0^-thalassemia and deletional α^+^-thalassemia from single *HBA* gene deletions (genotype --/-α). Most cases have mild anemia, with hemoglobin levels of 9–10 g/dL [3]. On the other hand, non-deletional Hb H disease resulting from compound heterozygosity of α^0^-thalassemia and a point mutation on one *HBA* gene (genotype --/α^T^α or --/αα^T^), such as Hb H/Constant Spring (--/α^CS^α) or Hb H/Pakse disease (--/α^PS^α), is generally associated with more severe symptoms. During the hemolytic crisis, which is frequently related to acute infections with high fever, the Hb level may drop significantly, leading to shock or renal shutdown. Some cases of non-deletional Hb H disease are severe enough to develop hydrops fetalis [4,5]. These fetuses can be monitored for early detection of severe anemia that requires blood transfusion in utero [6].

Currently, studies on fetal life with Hb H disease have rarely been published, in spite of the fact that the natural course of the disease is much more severe during fetal life than in postnatal life [4,7]. Hb H disease is also associated with hydrops fetalis. The course of the disease tends to be less severe after birth because of the switch of the globin chain production from predominantly gamma-chains to predominantly beta-chains, leading to an increasing amount of Hb H (β4) replacing Hb Barts (γ4). Newborns and adult patients with Hb H disease may have passed through a critical period of anemic hypoxia during fetal life without proper care or intrauterine treatment. Overwhelming evidence supports the notion that survivors of hypoxia in utero are commonly vulnerable to several adult diseases, especially to cardiovascular disorders, known as fetal programming or fetal origin of adult diseases. Convincingly, several patients with Hb H disease have poor quality of life associated with hypoxic cellular damage in developing vital organs in the prenatal period. Because fetal hypoxia caused by Hb H disease could be prevented or treated in utero to avoid residual insults in postnatal life, prenatal diagnosis of Hb H disease is increasingly performed and should be encouraged, especially in geographical areas of high prevalence, such as Thailand and Southeast Asia. 

Therefore, the screening for traits of α^0^-thalassemia (α^0^-thal) and α^+^-thalassemia (α^+^-thal) to identify the couples at risk of fetal Hb H disease is needed. In daily practice, we routinely screen for pregnancies at risk of fetal Hb Barts disease (homozygous α^0^-thal) using the simple mean corpuscular volume (MCV) as a simple screening test and polymerase chain reaction (PCR)-based methods as the confirmation technique. Unfortunately, the screening of the α^+^-thal trait using the MCV is not yet satisfactory enough because the MCV of the α^+^-thal trait is not much lower than that of normal individuals, unlike the much smaller MCV of the α^0^-thal trait. As a result, the combination of all red blood cell indices (called RBC indices) obtained from clinical practice might be able to improve the accuracy of screening the α^+^-thal trait. In the meantime, machine learning (ML)-based methods that are capable of accurately predicting the α^+^-thal trait without the use of any experimental evidence play a crucial role in facilitating the large-scale prediction of the α^+^-thal trait.

## 2. Materials and Methods

### 2.1. Study Subjects

The secondary analysis of a prospective database was conducted at Maharaj Nakorn Chiang Mai Hospital (a tertiary center and medical school), Chiang Mai University, Thailand, from July 2021 to March 2022. The study was reviewed and approved by the institutional review board (Research Ethics Committee 4, Faculty of Medicine, Chiang Mai University, study code OBG-2564-08545). The data were retrieved anonymously from the primary study of prenatal surveillance of Hb H disease to obtain RBC parameter information from molecularly confirmed α^+^-thal traits and their normal counterparts. The study population was pregnant women and their husbands attending the antenatal care clinic at our hospital and network hospitals who were a possible couple at risk for fetus with Hb H disease. The inclusion criteria were as follows: (1) pregnant woman or her husband (partner) attending antenatal care for the first time regardless of gestational age; (2) healthy and without any hematological disease; (3) at least one of the couple had a mean corpuscular volume (MCV) < 80 fL from the screening blood count. The exclusion criteria were as follows: (1) multifetal pregnancy, (2) one or both of the couple were anemic (Hb of less than 10 g/dL) due to causes other than thalassemia. The subjects were recruited with written informed consent. Note that only normal and alpha^+^-thalassemia groups were included in the analysis.

Steps of the research procedure: In the primary study, the couples attending the antenatal care clinic were counseled and invited to participate. The participants provided written informed consent. The baseline characteristics of the participants (age, gender, history of anemia) were assessed and recorded. Four milliliters of blood samples were collected from the participants. The blood samples were analyzed for (1) the complete blood count (CBC) using an automated hematology analyzer to determine the levels of hemoglobin (Hb), hematocrit (Hct), mean corpuscular volume or mean cell volume (MCV), mean corpuscular hemoglobin (MCH), mean corpuscular hemoglobin concentration (MCHC), RBC distribution width (RDW), and red blood cell (RBC) count; (2) the hemoglobin analysis by high-pressure liquid column chromatography (HPLC) using a Variant II HPLC system (Bio-Rad Laboratories, Berkeley, CA, USA); and (3) the molecular diagnosis of α^0^-thalassemia (SEA, Thai deletion), α^+^-thalassemia (3.7 kb deletion, 4.2 kb deletion), Hb CS and Hb Pakse mutations using standard polymerase chain reaction (PCR)-based methods. The participants were treated as per the standard antenatal care. Definitive diagnosis of thalassemia status of the participants was based on the Hb analysis and molecular diagnosis results. The cases of alpha-thalassemia disease (Hb H disease), Hb variants other than Hb Constant Spring (CS) and Hb Pakse, beta-thalassemia trait/disease, Hb E trait/disease were excluded at this step. 

The cases were classified into three categories for the analysis purposes:Normal (not thalassemia): cases with normal Hb analysis results (Hb AA_2_ pattern, A_2_ ≤ 3.5%) and negative PCR results for α^0^-thalassemia, α^+^-thalassemia, Hb CS and Hb Pakse mutations;α^+^-Thalassemia trait: heterozygous α^+^-thalassemia, Hb CS trait, Hb Pakse trait;Two-allele α-thalassemia mutation: heterozygous α^0^-thalassemia, homozygous α^+^-thalassemia, and α^+^-thalassemia/Hb CS or α^+^-thalassemia/Hb Pakse.

### 2.2. Sample Size

According to our pilot study on the diagnostic performance of the model, the sensitivity of red blood cell indices in predicting the α^+^-thalassemia trait is approximately 75%. To gain a power of test of 80% at the 95% confidence interval with an allowable error of 0.10, the study needed at least 50 cases of the α^+^-thal trait.

### 2.3. Statistical Analysis

Continuous variables including age and red blood cell parameters were expressed as the means ± standard deviation and were compared between groups using Student’s *t*-test. The frequency of gender was reported as the number and percentage and compared using the chi-squared test. The statistical analysis was performed using SPSS Statistics for Windows, version 22.0 (IBM Corporation, Armonk, NY, USA). The *p*-value < 0.05 was considered significant.

#### 2.3.1. Conventional Statistical Analysis

The receiver operator characteristic (ROC) curve was created to show the diagnostic performance of red blood cell parameters in predicting the α^+^-thalassemia trait against normal controls. Red blood cell parameters were used as a single parameter and a combination model derived from a conventional binary logistic regression analysis.

#### 2.3.2. Model Construction and Development

In this study, we developed variant ML-based models by using 11 different ML methods (i.e., convolutional neural networks (CNNs), support vector machine (SVM), multilayer perceptron (MLP), random forest (RF), partial least squares (PLS), logistic regression (LR), extremely randomized trees (ET), light gradient boosting machine (LGBM), extreme gradient boosting (XGB), decision tree (DT), k-nearest neighbors (KNNs)). Herein, we performed a 10-fold cross-validation test in order to optimize and create all the ML-based models by using the Scikit-learn v0.22 package [8] with the 10-fold cross-validation test, while we selected the best-performing model in terms of cross-validation accuracy (ACC) for the final model construction. In addition, to assess the generalization capability and robustness of the models, we evaluated each model based on an independent test. More details on ML-based model construction are provided in our previous studies [9,10,11].

#### 2.3.3. Performance Evaluation

As seen in Table 1, the numbers of α^+^-thal trait and normal samples (referred to as the positives and negatives, respectively) were 160 and 229, respectively. Herein, around 80% of the positives and negatives were considered for constructing the training dataset, while the remaining samples were considered for constructing an independent test dataset. As a result, the training dataset consisted of 126 positives and 185 negatives, while the independent test dataset consisted of 34 positives and 44 negatives. To assess the predictive capability and robustness of the models, we utilized the means of five well-known performance measures, including ACC, sensitivity (Sn), specificity (Sp), Matthews correlation coefficient (MCC), and area under the ROC curves (AUC) [9,12]. ACC, Sn, Sp, and MCC are calculated as follows:ACC = (TP + TN)/((TP + TN + FP + FN)) (1)
Sn = TP/((TP + FN)) (2)
Sp = TN/((TN + FP)) (3)
MCC = (TP × TN − FP × FN)/√((TP + FP)(TP + FN)(TN + FP)(TN + FN)) (4)
where TN and TP are the numbers of true negatives and true positives, respectively. On the other hand, FN and FP are the numbers of false negatives and false positives, respectively [13,14,15].

## 3. Results

### 3.1. Baseline Data and Comparisons of the Hematologic Parameters

In total, 594 cases (female/male: 330/264) were included in the analysis. All the participants were of Thai nationality, and the mean age was 29.7 ± 6.6 years. There were 229 cases in the normal group, 160 cases with the α^+^-thalassemia trait, and 205 cases in the two-allele α-thalassemia mutation category, respectively. Their detailed diagnoses are shown in Table 1.

Regarding the comparison of red blood cell indices between the normal group and the α^+^-thalassemia trait group, the levels of Hb, MCV, MCH, and MCHC were significantly lower in the α^+^-thalassemia trait group, whereas the RDW and RBC count were significantly higher in the α^+^-thalassemia trait group, as presented in Table 2. Likewise, comparisons of red blood cell parameters between the control group and the two-allele mutation group, Hb, MCV, MCH, and MCHC were significantly lower in the two-allele mutation group, whereas the RDW and RBC count were significantly higher. Furthermore, it is noteworthy that all parameters but Hct in the α^+^-thalassemia trait group were significantly different from those in the two-allele mutation group, as presented in Table 2. In the two-allele mutation group, hematologic parameters had a more pronounced difference from the control than those in the α^+^-thalassemia trait group.

### 3.2. Performance Evaluation of Conventional Models

Based on ROC curves, the diagnostic performance of each red blood cell parameter and their combination in predicting the α^+^-thalassemia trait is presented in Figure 1 and Table 3. The MCH was the best parameter in predicting the α^+^-thal trait, with an AUC of 0.857, and a slightly better performance was observed when it was combined with the MCV, MCHC, and Hb. In the prediction of the α^+^-thal trait, the combined parameters derived from the binary logistic regression analysis provided an AUC of 0.868, giving a sensitivity of 80% at a false positive rate of 25%. Therefore, red blood cell indices have a relatively good diagnostic performance regarding the α^+^-thal trait.

### 3.3. Performance Evaluation of Different ML-Based Models

In this section, we conducted a comparative analysis of various ML-based models (i.e., CNNs, DT, ET, KNNs, LGBM, LR, MLP, PLS, RF, SVM, and XGB) trained with RBC indices in α^+^-thal trait prediction. Specifically, we performed a 10-fold cross-validation scheme for creating each ML-based model, and an independent test was then used for evaluating the generalization capability and robustness of the models. As mentioned above, the best-performing model in terms of cross-validation ACC was selected for the final model construction. The performance of nine ML-based models is recorded in Table 4 and Table 5. As can be seen from Table 4, the top three powerful ML-based models in α^+^-thal trait prediction contain CNNs or DL (DeepThal) (specifically named as the best model in this study), SVM, and MLP with a corresponding ACC of 80.69%, 80.68%, and 79.72, respectively. The DL (DeepThal) model gave an Sn and Sp of 80% and 81%, respectively. For convenience of comparison, we considered these top three powerful models for further analysis (Figure 2 and Figure 3). Overall, CNNs outperformed other ML-based models in terms of ACC, Sn, and MCC on the training dataset. In case of the performance of the independent test, CNNs still attained the best performance in terms of ACC, Sn, and MCC, which were 5.13, 11.77, and 0.104, respectively, higher than those of the second-best model (SVM) (Table 5). Thus, the CNN model coupled with all the red blood cell indices is referred to herein as DeepThal. Note that DeepThal refers to the deep learning model derived by the machine learning procedure using a model of convolutional neural networks (CNNs) based on various routine CBC hematological parameters specifically created to predict the risk of having the alpha^+^-thalassemia trait among a healthy population.

### 3.4. Mechanistic Interpretation of DeepThal

Here, we employed the Shapley Additive exPlanation (SHAP) method [16] for elucidating the contribution of each feature to the prediction outcomes and highlighting the most important feature. Figure 3 shows the SHAP values of all the RBC indices, where positive and negative SHAP values represent the α^+^-thal trait and control predictions, respectively. As can be seen, the top three important features for DeepThal consisted of the MHC, MCV, and sex. The SHAP values of the MHC and MCV were relatively low for most of the α^+^-thal trait patients and high for most of the control patients. As a result, given an unknown patient, low values of these two indices indicate that the patient is predicted as α^+^-thal trait; otherwise, the patient is predicted as a control case.

## 4. Discussion

In this study, we developed DeepThal, a deep learning (DL) framework for the large-scale prediction of α^+^-thalassemia trait using age, gender, and the red blood cell parameters. The major contributions of DeepThal are summarized in the following three main aspects: (i) to the best of our knowledge, DeepThal is the first DL-based model for α^+^-thalassemia trait prediction; (ii) we systematically investigated the predictive capability of various ML-based models for α^+^-thalassemia trait prediction in terms of both 10-fold cross-validation and independent tests; and (iii) the experimental results indicated that DeepThal achieved a better performance compared with other ML-based methods in terms of the independent test dataset, with an ACC of 80.77%, Sn of 70.59%, and MCC of 0.608.

Routine screening for α^0^-thal with the MCV or the osmotic fragility test is the standard practice of our antenatal care for prenatal diagnosis of Hb Barts disease [17,18], whereas α^+^-thal has never been screened for in clinical practice since in the past Hb H disease was not targeted for prenatal diagnosis because it is nonlethal, and intrauterine treatment was not available. Nevertheless, as mentioned earlier, prenatal diagnosis and intrauterine treatment of Hb H disease in cases of severe anemia is more commonly practiced. However, effective screening for the α^+^-thal trait to identify couples at risk of Hb H disease has rarely been described. Only few studies on α^+^-thal trait screening have been published and are summarized as follows: Jindadamrongwech et al. [19] demonstrated that the red blood cell parameters of patients with the α^+^-thal or Hb Constant Spring trait are not significantly different from those of normal individuals. Anselmo et al. [20] demonstrated that patients with the α^+^-thal trait have normal levels of Hct (43.13 ± 3.03%), but the levels of Hb, MCV, MCH, MCHC, and RDW significantly differ from normal ones. However, the use of these parameters in predicting the α^+^-thal trait was not analyzed. Tayapiwatana et al. [21] developed a test kit to screen for alpha-thalassemia using the immunochromatographic strip test and showed that the kit has a sensitivity of 82.9% in the detection of the α^+^-thal trait. Finally, Makonkawkeyoon et al. [22] developed a test kit to screen for alpha-thalassemia, a simple ELISA strip test using the monoclonal antibody specific to Hb Barts. The test was used to screen 135 patients or traits from 39 families, using molecular genetic diagnosis as a gold standard. The authors showed that the ELISA strip test could detect Hb H (--/-α) 100% of the time (49/49); α^0^-thalassemia (--/αα)—100% (51/51); α^+^-thal (−3.7 kb; -α3.7/αα)—76.9% (30/39); α^+^-thal (−4.2 kb; -α4.2/αα)—100% (2/2); and heterozygous Hb CS (αCSα/αα)—92.3% (24/26). Accordingly, the ELISA strip test is not highly sensitive for α^+^-thal (-α3.7: the common type) (76.9%). The definite diagnosis of α^+^-thal can be made using a PCR technique [23]. However, this method is expensive, needs expertise, and is not widely available for use. Thus, the PCR is not appropriate as the primary screening test for the α^+^-thal trait.

According to our findings and previous studies, single parameters of red blood cell indices seem to be less effective in screening for the α^+^-thal trait because of a too-low detection rate and high false positive rate, implying a great number of missed diagnoses and unnecessary molecular genetic confirmation. Of those parameters, the MCH seems to be most predictive, giving an AUC of 0.718. The performance is better when a conventional logistic predictive model is used, giving an AUC of 0.868. However, with the conventional model, to achieve a detection rate of 80%, the false positive rate is still unacceptable, as high as 25%. The ML-derived model is superior to the conventional model, giving an Sn and Sp of 80% and 81%, respectively. This study suggests that the ML-derived model of red blood cell indices has an acceptable performance in predicting the α^+^-thal trait, comparable with that of the ELISA strip test [22] and the immunochromatographic strip test [21]. However, the most important advantage of the model using red blood cell indices is that it needs no extra tests other than the blood test routinely performed at antenatal care clinics.

The limitations of this study are as follows: (1) Since the performance of the model in this study is derived from the selected population with a high prevalence of the alpha^+^-thalassemia trait, its ability in prediction might be different when applied to other groups of population with low prevalence. Nevertheless, the performance should not be much different since sensitivity and specificity are typically similar between different populations, whereas predictive values vary with the prevalence. Accordingly, though we developed the model from the selected population, similar performance should be expected when used in other groups. (2) The sample size was relatively small. Though in this preliminary study DeepThal seems to be useful as a screening test, the best cutoff probability to be used as a criterion for confirmatory tests is yet to be determined in future studies based on prevalence and cost-effective analysis.

## 5. Conclusions

In conclusion, among the red blood cell indices, the MCH, as a single parameter, has the highest performance in predicting the α^+^-thal trait, whereas the AUC is relatively low in predicting the α^+^-thal trait. The ML-derived model improves the diagnostic performance, giving an Sn and Sp of 80% and 81%, respectively, which may be acceptable for clinical practice. DeepThal achieved a better performance compared with other ML-based methods in terms of the independent test dataset, with an ACC of 80.77%, Sn of 70.59%, and MCC of 0.608.

Finally, our findings imply that our model is likely helpful in identifying couples at risk and prenatal diagnosis of Hb H disease, especially in the geographical areas of high prevalence. However, our results are based on a limited sample size, though the performance seemed to be reproducible when the model was tested on an independent test dataset. Therefore, the performance of the model should be further tested in other groups of population and with a larger sample size to determine whether it is reproducible or not. If the model performance is confirmed, we can take advantage of routine laboratory CBC tests at antenatal care clinics to evaluate the risk of fetal Hb H disease. In practice, our DeepThal model could be further developed into a user-friendly application for ease of use in daily practice.

## Figures and Tables

**Figure 1 jcm-11-06305-f001:**
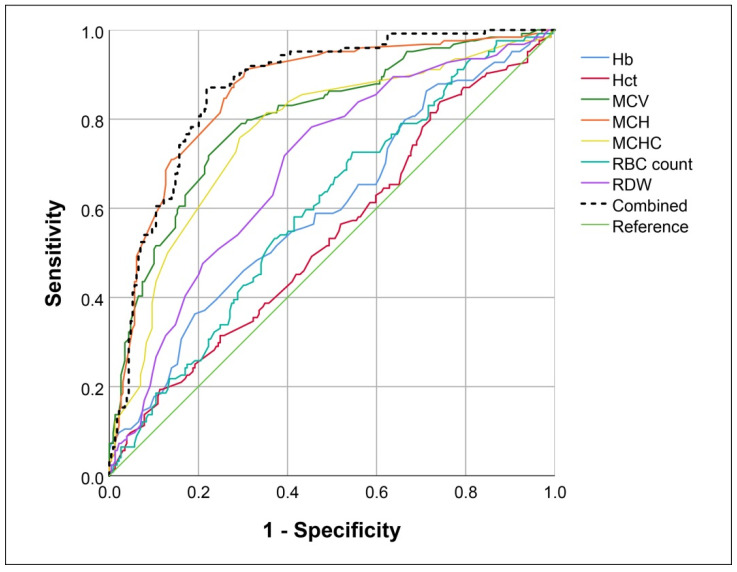
ROC curves show the diagnostic performance of red blood cell indices in predicting the α^+^-thalassemia trait. Hb, hemoglobin; Hct, hematocrit; MCH, mean corpuscular hemoglobin; MCHC, mean corpuscular hemoglobin concentration; MCV, mean corpuscular volume; RBC, red blood cell; RDW, red blood cell distribution width.

**Figure 2 jcm-11-06305-f002:**
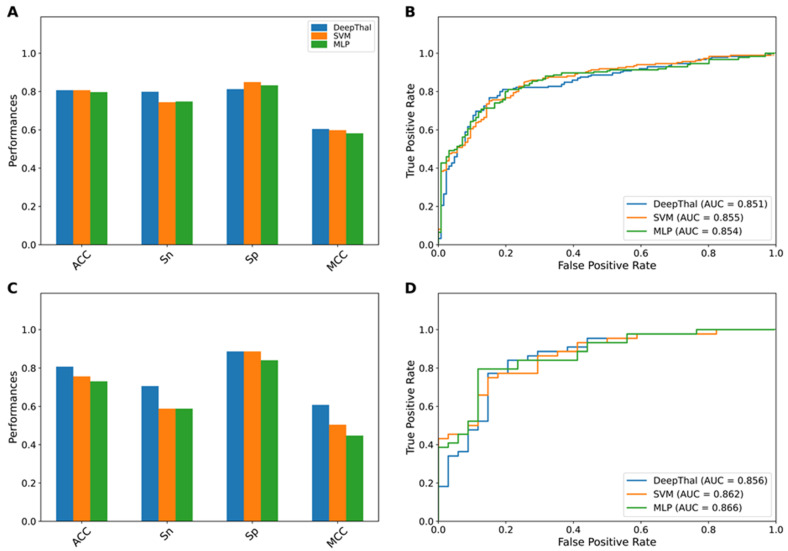
Performance comparison of the top three ML-based models on the training (**A**,**B**) and independent test (**C**,**D**) datasets. Prediction results of the top three ML-based models in terms of ACC, Sn, Sp, and MCC (**A**,**C**). ROC curves and AUC values of the top three ML-based models (**B**,**D**). SVM, support vector machine; MLP, multilayer perceptron; ACC, accuracy; AUC, area under the curve; MCC, Matthews correlation coefficient; ML, machine learning; Sn, sensitivity; Sp, specificity.

**Figure 3 jcm-11-06305-f003:**
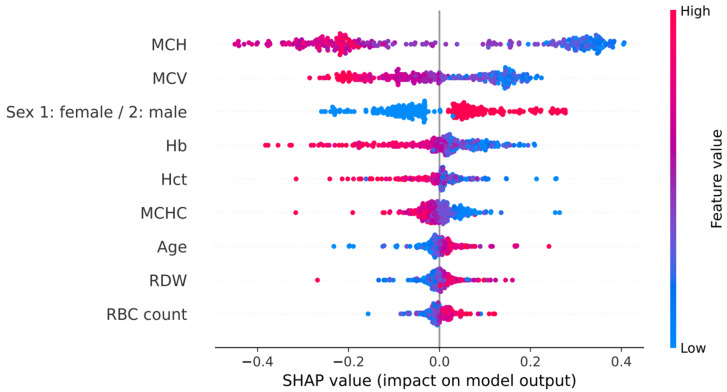
SHAP values of the nine red blood cell indices employed by the proposed DeepThal, where positive and negative SHAP values represent the α^+^-thal trait and control predictions, respectively. Hb, hemoglobin; Hct, hematocrit; MCH, mean corpuscular hemoglobin; MCHC, mean corpuscular hemoglobin concentration; MCV, mean corpuscular volume; RBC, red blood cell; RDW, red blood cell distribution width.

**Table 1 jcm-11-06305-t001:** Distribution of the α-thalassemia status.

α-Thalassemia Status	Group	Frequency	Percentage
Normal (not thalassemia trait)	Normal	229	38.5
α^+^-thalassemia trait	α^+^-thal trait	124	20.9
Hb Constant Spring trait		36	6.1
Homozygous α^+^-thalassemia	Two-allele α-thal mutation	5	0.8
α^+^-thalassemia trait/Hb Constant Spring trait		3	0.5
α^0^-thalassemia trait		197	33.2
**Total**		**594**	**100.0**

**Table 2 jcm-11-06305-t002:** Comparisons of clinical and red blood cell parameters between the α^+^-thalassemia trait (α^+^-thal trait) vs. normal controls (not thalassemia), two-allele α-thal mutation vs. normal controls, and α^+^-thal trait vs. two-allele α-thal mutation trait.

Clinical and Red Blood Cell Parameters	Control	α^+^-Thal Trait	*p*-Value Control vs. α^+^-Thal Trait	Two-Allele α-Thal Mutation	*p*-Value Control vs. Two-Allele α-Thal Mutation	*p*-Value Two-Allele α-Thal Mutation vs. α^+^-Thal Trait
Age	29.8 ± 6.8	29.3 ± 6.1	0.531	30.2 ± 6.6	0.525	0.235
Female/male	121/108	69/55	0.616	121/84	0.252	0.615
Hb (g/dL)	13.7 ± 1.7	13.0 ± 1.7	0.002	12.3 ± 1.4	<0.001	<0.001
Hct (%)	40.4 ± 5.1	39.7 ± 5.1	0.206	39.3 ± 4.8	0.046	0.571
MCV (fL)	86.7 ± 4.3	81.6 ± 4.2	<0.001	67.2 ± 4.2	<0.001	<0.001
MCH (pg)	29.4 ± 1.8	26.9 ± 1.7	<0.001	21.1 ± 1.6	<0.001	<0.001
MCHC (g/dL)	34.0 ± 2.8	32.9 ± 1.1	<0.001	31.4 ± 1.1	<0.001	<0.001
RDW (%)	12.9 ± 0.9	13.5 ± 1.0	<0.001	16.7 ± 2.0	<0.001	<0.001
RBC count (×10^6^/mm^3^)	4.7 ± 0.6	4.9 ± 0.6	0.002	5.9 ± 1.3	<0.001	<0.001

Hb, hemoglobin; Hct, hematocrit; MCH, mean corpuscular hemoglobin; MCHC, mean corpuscular hemoglobin concentration; MCV, mean corpuscular volume; RBC, red blood cell; RDW, red blood cell distribution width.

**Table 3 jcm-11-06305-t003:** Performance of red blood cell parameters in predicting the α^+^-thalassemia trait.

Red Blood Cell Parameters	AUC (95% CI)	Cutoff	Sn (%)	Sp (%)
Hb (g/dL)	0.600 (0.538–0.663)	12.15	36.3	80.8
Hct (%)	0.539 (0.476–0.602)	44.95	83.9	25.8
MCV (fL)	0.801 (0.753–0.850)	83.95	70.2	77.7
MCH (pg)	0.857 (0.816–0.899)	28.95	78.1	70.4
MCHC (g/dL)	0.767 (0.714–0.820)	33.3	75.8	70.7
RBC count	0.599 (0.539–0.660)	4.50	72.6	45.4
RDW (%)	0.692 (0.636–0.749)	12.95	78.2	54.6
Combined parameters	0.868 (0.830–0.906)	0.316	80.1	75.1

AUC, area under the curve; CI, confidence interval; Hb, hemoglobin; Hct, hematocrit; MCH, mean corpuscular hemoglobin; MCHC, mean corpuscular hemoglobin concentration; MCV, mean corpuscular volume; RBC, red blood cell; RDW, red blood cell distribution width; Sn, sensitivity; Sp, specificity.

**Table 4 jcm-11-06305-t004:** Cross-validation results of different ML-based predictors in the training dataset.

Method	ACC (%)	Sn (%)	Sp (%)	MCC	AUC
DL (DeepThal) *	**80.69**	**79.87**	81.23	**0.604**	0.851
SVM	80.68	74.40	84.94	0.598	**0.855**
MLP	79.72	74.77	83.27	0.581	0.854
RF	79.41	75.73	82.16	0.577	0.857
PLS	79.39	70.31	85.48	0.567	0.846
LR	79.07	68.44	**85.87**	0.556	0.854
ET	78.15	73.81	81.16	0.549	0.831
LGBM	78.11	74.35	80.96	0.549	0.842
XGB	77.79	76.94	78.28	0.547	0.840
DT	72.33	70.63	73.16	0.433	0.719
KNNs	69.43	62.58	74.09	0.367	0.683

* DL (DeepThal) refers to an ML-based model using convolutional neural networks (CNNs). The best performance value for each metric is highlighted in bold. ACC, accuracy; AUC, area under the curve; MCC, Matthews correlation coefficient; ML, machine learning; Sn, sensitivity; Sp, specificity.

**Table 5 jcm-11-06305-t005:** Independent test results of different ML-based predictors in the independent test dataset.

Method	ACC (%)	Sn (%)	Sp (%)	MCC	AUC
DL (DeepThal) *	**80.77**	**70.59**	88.64	**0.608**	0.856
RF	78.21	**70.59**	84.09	0.554	0.838
LR	76.92	58.82	**90.91**	0.534	0.860
PLS	76.92	58.82	**90.91**	0.534	**0.865**
SVM	75.64	58.82	88.64	0.504	0.862
ET	75.64	64.71	84.09	0.501	0.849
LGBM	75.64	61.76	86.36	0.502	0.824
XGB	74.36	58.82	86.36	0.475	0.815
MLP	73.08	58.82	84.09	0.447	0.866
DT	69.23	47.06	86.36	0.368	0.667
KNNs	67.95	52.94	79.55	0.339	0.662

* DL (DeepThal) refers to an ML-based model using convolutional neural networks (CNNs). The best performance value for each metric is highlighted in bold. ACC, accuracy; AUC, area under the curve; MCC, Matthews correlation coefficient; ML, machine learning; Sn, sensitivity; Sp, specificity.

## Data Availability

The datasets analyzed during the current study are available from the corresponding author upon reasonable request.

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
