# Peer review of "DeepThal: A Deep Learning-Based Framework for the Large-Scale Prediction of the α+-Thalassemia Trait Using Red Blood Cell Parameters"

_jcm, 2022, doi:10.3390/jcm11216305_

Round 1
Reviewer 1 Report
The authors developed several machine learning-based models for predicting a+-thalassemia among the possible couples at risk for fetus with Hb H disease. Based on the performance tested on the independent test dataset, they proposed the “Deep Thal” as the best prediction model.
The paper is technically sound and well-written in a logical way. However, there are a few concerns as follows;
Considering that the performance of the ML-based models was tested among the selected population (the couples at risk for fetus with Hb H disease), the findings are therefore limited to this selected group. The exact implications of the findings and/or how the “Deep Thal” facilitates the large-scale prediction of a+-thal trait should be discussed. Also, limitations of the study should be given.
Please state clearly whether only normal and a+-thal groups were included in the statistical analysis of performance. This information is crucial for determining the reproducibility and the applicability of the results.
Other minor issues:
Since the term “Deep Thal” was invented by the authors and it is the key point of the paper, a clearer description should be provided.
There were some inconsistencies:
Page 7/12, lines 233 mentioned that CNN was one of the top 3 models, but no CNN shown in Table 4.
Tables 4 and 5; please make the tables complete by providing a brief explanation (for DL) at the bottom.
Page 7/12, lines 239; figures “5.13, 11.77, and 10.40” cannot be traced back (Table 5).
Author Response
Reviewer #1: (response in red)
Comments and Suggestions for Authors
The authors developed several machine learning-based models for predicting a+-thalassemia among the possible couples at risk for fetus with Hb H disease. Based on the performance tested on the independent test dataset, they proposed the “Deep Thal” as the best prediction model.
The paper is technically sound and well-written in a logical way. However, there are a few concerns as follows;
Considering that the performance of the ML-based models was tested among the selected population (the couples at risk for fetus with Hb H disease), the findings are therefore limited to this selected group. The exact implications of the findings and/or how the “Deep Thal” facilitates the large-scale prediction of a+-thal trait should be discussed. Also, limitations of the study should be given.
Response:
- Concerning studying in the selected group, we add the comments on this issue in “Discussion”, as highlighted in the revised MS, described as a limitation, at the paragraph just before the conclusion. Also part of limitation is added.
- Concerning its implications, we add the comments at the end of “Conclusion”, as highlighted. In the revised MS.
Please state clearly whether only normal and a+-thal groups were included in the statistical analysis of performance. This information is crucial for determining the reproducibility and the applicability of the results.
Response: In the revised MS, we clearly state in “Methods” that only normal and alpha+-thal groups were included, as highlighted at the end of the first paragraph of page 3. Because alpha0-thal group and other anemic groups can usually be screened by MCV in practice and cases with anemia are always worked up for the causes. So they were not included in study.
Other minor issues:
Since the term “Deep Thal” was invented by the authors and it is the key point of the paper, a clearer description should be provided.
Response: In the revised MS, the description of “DeepThal” is elaborated at the end of page 7, as highlighted, as follows:
DeepThal refers to the deep learning model derived by machine learning procedure, using model of convolutional neural networks (CNN), based on various hematological parameters of routine CBC lab, specifically created to predict the risk of having alpha+-thalassemia trait among healthy population.
There were some inconsistencies:
Page 7/12, lines 233 mentioned that CNN was one of the top 3 models, but no CNN shown in Table 4.
Response: Lines 233, CNN is the same as DL (DeepThal), which is specifically named for the best model in this study. In the revised MS, DL (DeepThal) is added to indicate the same as “CNN”.
Tables 4 and 5; please make the tables complete by providing a brief explanation (for DL) at the bottom.
Response: The brief explanation of DL (DeepThal) is added at the footnote of Table 4 and Table 5, as highlighted.
Page 7/12, lines 239; figures “5.13, 11.77, and 10.40” cannot be traced back (Table 5).
Response: DL (DeepThal) or CNN has an ACC, Sn and MCC 5.13, 11.77 and 0.104 higher than that of SVM. In the revised MS, we have change 10.40% to be 0.104 to be more consistent with that in Table 5 (the figures indicate the difference between DL (DeepThal) and SVM in Table 5).

Reviewer 2 Report
Thank you for sharing this research. The ability to screen for alpha thalassemia based on the CBC is indeed valuable for potential parents and for preventative medicine.
What is the lower age limit for using DeepThal? Reference ranges for CBC parameters vary with age so that would affect your cutoffs and potentially sensitivity/specifity.
If this model were used in the future, what criteria would you use to determine whether confirmatory studies (electrophoresis and/or PCR) are needed? Does the DeepThal test give a "confidence" measure of its conclusion?
Author Response
Reviewer #2: (response in blue)
Comments and Suggestions for Authors
Thank you for sharing this research. The ability to screen for alpha thalassemia based on the CBC is indeed valuable for potential parents and for preventative medicine.
What is the lower age limit for using DeepThal? Reference ranges for CBC parameters vary with age so that would affect your cutoffs and potentially sensitivity/specifity.
Response: There is no need for using cut-off for all parameters. The machine learning uses all significant parameters to learn and understand and automatically adjust for prediction.
If this model were used in the future, what criteria would you use to determine whether confirmatory studies (electrophoresis and/or PCR) are needed? Does the DeepThal test give a "confidence" measure of its conclusion?
Response: We added the comment on this issue in “Discussion”, as highlighted in the part of limitations (item 2) in the paragraph just before the “Conclusion”.
